# Gamma Radiation: An Eco-Friendly Control Method for the Rice Weevil, *Sitophilus oryzae* (L.) (Coleoptera: Curculionidae)

**DOI:** 10.3390/biology11091295

**Published:** 2022-08-31

**Authors:** George M. Nasr, El-Kazafy A. Taha, Amal M. Hamza, Eslam A. Negm, Nevein L. Eryan, Ahmed Noureldeen, Hadeer Darwish, Mohamed S. Zayed, El-Said M. Elnabawy

**Affiliations:** 1Agricultural Research Center, Plant Protection Research Institute, Stored Product Pests Department, Sakha, Kafrelsheikh 12611, Egypt; 2Department of Economic Entomology, Faculty of Agriculture, Kafrelsheikh University, Kafrelsheikh 33516, Egypt; 3Agricultural Research Center, Field Crops Research Institute, Crop Physiological Research Department, Sakha, Kafrelsheikh 12611, Egypt; 4Department of Biology, College of Science, Taif University, P.O. Box 11099, Taif 21944, Saudi Arabia; 5Biotechnology Department, College of Sciences, Taif University, P.O. Box 11099, Taif 21944, Saudi Arabia; 6Department of Pesticides, Faculty of Agriculture, Damietta University, Damietta 34517, Egypt

**Keywords:** gamma irradiation, mortality, sterility, chlorophyll, proline, progeny, germination

## Abstract

**Simple Summary:**

This study was conducted to evaluate the effect of gamma radiation at a dosage of 0.10, 0.25, 0.50, and 1.00 kGy on the mortality percentages of *Sitophilus oryzae* adults, their effects on weevil sterility, and seed germination. The dosage of 1.00 kGy caused 100% mortality after 96 h of irradiation. Moreover, the use of irradiation at a dosage of 1.00 kGy against *S. oryzae* adults has strong damage on histological alterations. It also has a negative impact on the germination percentage. Thus, the use of gamma radiation is a very important alternative method to protect stored grains and their products against rice weevils. On the contrary, it is not recommended to use irradiated grains for farming.

**Abstract:**

The rice weevil, *Sitophilus oryzae* (L.), is the most destructive insect pest of stored cereals worldwide. The current study was conducted to determine the lethal, reproductive, and histological effects of gamma irradiation on *S. oryzae* adults. In addition, the impact on germination, chlorophyll, and proline content in wheat seedlings from treated grains was determined. Wheat grains were infested with rice weevil adults and then irradiated by gamma rays. Gamma radiation was applied at a dosage of 0.10, 0.25, 0.50, and 1.00 kGy. Mortality percentage and LD_50_ were recorded after 48, 72, 96, and 120 h of treatment. The dosage of 1.00 kGy caused 100% mortality after 96 h of irradiation. The required dosage of gamma radiation to kill 50% (LD_50_) of adults after 48 h was 1.51 kGy. All tested doses caused complete sterility to 24 h old adults. A histological alteration was noticed at a dosage of 1.00 kGy, which showed cytoplasmic vacuolization, tissues exhibiting signs of putrefaction, and necrosis of cells; furthermore, gamma irradiation affected chlorophyll a and b. The highest amounts were detected in wheat seedlings from grains irradiated at 0.10 kGy. There was a significant increase in plant proline content at the higher doses (0.50 and 1.00 kGy) compared with seedlings from nonirradiated grains. It could be concluded that gamma radiation can be used as an eco-friendly trend to control stored-product pests without any residual effects.

## 1. Introduction

Some insect pests are the main problem of stored grains worldwide because they can significantly decrease the quantity and quality of grain and its products [1,2]. The rice weevil, *S. oryzae* (L.) (Coleoptera: Curculionidae), is the most widespread and destructive major insect pest of stored rice, other cereals, and their products worldwide and drastically decreases yields [3]. Control of this pest relies on the widespread use of various residual synthetic insecticides and fumigation with methyl bromide or hydrogen phosphide. However, methyl bromide was classified as an ozone-depleting substance, and its use was recently banned in many countries [4]. Increased public concern over the residual toxicity of insecticides applied to stored grain, the occurrence of insecticide-resistant insect strains [5], toxicity to nontarget organisms [6], and the precautions necessary to work with traditional chemical insecticides calls for alternative stored-product insect pest control methods. 

Conventional insecticides have many negative effects, such as uncontrolled release into the environmental system [7]. Recently, some researchers have been focusing on nontraditional control methods, such as using plant inducers to enhance plant systemic resistance against insect pests [8], as well as using some plant essential oils for controlling *Tribolium castaneum* [9]. Furthermore, El-Nabawy et al. [10,11] demonstrated that using flowering plants and organic fertilizers enhanced natural enemies, which suppressed some associated insect pests. Shawer et al. [12] and Taha et al. [13] studied the effect of emergence time and cold storage durations on the fitness aspects of *Trichogramma Evanecsens* (Westwood).

In recent years, many researchers have demonstrated that ionizing irradiation, especially X-rays and gamma irradiation, can effectively control stored grain pests without detrimental effects on the commodity [14,15]. Gamma irradiation is safe and environmentally friendly [16,17] and is approved for stored product use in at least 33 countries. Gamma irradiation was studied for its lethal effect, as well as for inhibiting the reproduction of many stored grain pest species [18,19]. For instance, Tuncbilek et al. [20] found that 0.10 kGy was the effective dose against both larvae and adults of *T. castaneum*, and LD_50_ and LD_99_ values were determined as 19.75 and 42.97 Gy for larvae and 33.21 and 64.50 Gy for the adult stage. According to Salim et al. [15], a 0.50 kGy dose of gamma radiation can kill 100% of *T. confusum* adults and larvae in 22 days. Meanwhile, Abbas and Nouraddin [21] reported that the dose of 0.20 kGy caused 100% mortality in irradiated *T. castaneum* adults 28 days after treatment. A generic dose of 0.60 kGy has been efficacious for controlling the Angoumois grain moth, *Sitotroga cerealella*, and the Indian meal moth, *Plodia interpunctella* [14]. In insects, gamma irradiation affects sites of ongoing cell division, which in the adult insect include the gonads and midgut. For example, exposure to an accumulated dose of 0.20 Gy gamma rays (0.66 rad/s dose rate) from the Cs137 source induces cellular perturbations in the midgut epithelium of the F1 progeny of the beetle *Blaps polycresta* [22]. At minimal gamma irradiation doses, which disrupt the functions of the digestive system, it was noticed that the midgut does not have a well-defined membrane, and the intercellular connection is decreased with adjacent cells [23]. The insects will not reproduce and will cease feeding because the midgut epithelium cannot process food. 

Gamma radiation interacts with atoms or molecules in plant cells, producing free radicals [24]. These radicals can modify or damage important substances of plant cells and have been shown to impose considerable effects on the physiology, morphology, or biochemistry of plants [25,26]. Radiation of seeds with high doses of gamma rays can disturb hormone balance, enzyme activity, photosynthetic pigment content, and synthesis of protein [27,28]. Many studies found a positive correlation between proline accumulation and enhancing stress tolerance in plants [29,30,31]. Moreover, AlKahtani et al. [32] reported that one of the most crucial factors for evaluating environmental stress on plants is the chlorophyll content of the leaves, which declines under pressure.

Our objective was to evaluate the effect of different doses of gamma irradiation on the mortality percentages and the progeny production of *S. oryzae*. This study was also designed to assess the effect of gamma irradiation (1.0 kGy) on the cellular organization and other tissues of *S. oryzae* adults. Additionally, we study the effects of gamma radiation on the germination, chlorophyll, and proline contents of wheat seedlings to assess the suitability of irradiated grains in farming. 

## 2. Materials and Methods

### 2.1. Mass Rearing of S. oryzae

Mass culture of *S. oryzae* was maintained at the laboratory of the Department of Stored-Product Pests, Plant Protection Research Institute, Sakha Agriculture Research Station, Kafrelsheikh, Egypt, without exposure to any insecticide. The colony was reared in a glass container and maintained under controlled laboratory conditions of 28 ± 1 °C, 65 ± 5% R.H, and continuous darkness. The insects were reared on clean wheat *Triticum vulgare* L. grains (variety Giza 171). All the experimental insects were newly emerged adults (1–7 days old).

### 2.2. Gamma Irradiation of S. oryzae Adults and Effects on Reproduction

Irradiation of *S. oryzae* was conducted at an irradiator facility with a 60Co gamma source at a dosage rate of 0.930 kGy/h, at the National Center for Radiation Research and Technology, Atomic Energy Authority, Cairo, Egypt. Radiation doses of 0.10, 0.25, 0.50, and 1.00 kGy were used. Fifty unsexed newly emerged adult weevils of *S. oryzae* were randomly selected from the stock culture by sieving and were transferred to Petri dishes containing 20 g of wheat (variety Giza 171) grains as food and then exposed to gamma irradiation with the previous concentrations. A control treatment was subjected to the same conditions without radiation exposure. Immediately after treatment, irradiated and nonirradiated weevils were returned to the laboratory, and Petri dishes were placed in an incubator at a temperature of 28 ± 1 °C and 65 ± 5% RH. Four replicates were used for each dose level and control. Adult weevil mortality was recorded after 48, 72, 96, and 120 h, and dead insects were removed. The median lethal dose (LD_50_) of gamma rays with a 95% confidence limit was estimated by probit analysis. All containers were returned to the incubator for 60 days. After 60 days of irradiation, all emerged *S. oryzae* adults were counted and recorded. 

### 2.3. Histological Technique

Irradiated (dose 1.00 kGy) and nonirradiated *S. oryzae* adults treated as described previously were fixed in 10% buffered formalin (formaldehyde) for about 24–48 h. After the fixation, the insects were dehydrated in an ascending grade of ethanol alcohol for 2 min (100: 70%) and embedded in paraffin wax. Eighty µm thick paraffin sections were cut on a Reichert ultramicrotome with a diamond knife (RMC PT-XL Power Tome Ultra microtome) and mounted on microscope slides, then subjected to hematoxylin–eosin staining. The slides were soaked in an ascending alcohol series (70, 80, 90, and 96) and placed in absolute alcohol for 10 min, then passed into two changes of xylol. Finally, the slides were fixed in Canada balsam and covered with cover glass and desiccated at 40 °C for a day. The sections were observed by a light microscope.

### 2.4. Effect of Gamma Irradiation on Germination

Wheat seeds were irradiated with different doses as mentioned above. The seeds are sterilized and placed in Petri dishes on filter paper containing each 10 mL water 20 seeds for each treatment. Each irradiation dose was repeated in three replicates and germinated at 20 ºC with an average 8 h photo phase, under laboratory conditions. Germination percentages were considered when they exhibited a radical extension of >2 mm. Germination seeds were scored daily for 7 days. The final germination percentage (FGP) was calculated according to the following equation [33]:FPG = (Nt × 100)/N
where Nt = the proportion of germination seeds in each treatment for measurement; N = the total number of seeds in the bioassay.

### 2.5. Determination of Chlorophyll and Proline Content of T. vulgare L. Seedlings

This phase was carried out to determine the effects of gamma radiation on the physiological characteristics of wheat seedlings during the growing seasons in 2019/2020 and 2020/2021. One kilogram of wheat grains (Giza 171) was selected for each irradiation dose. The nonirradiated grains served as control. The irradiated and nonirradiated grains were sown under natural growing conditions in the field of the Experimental Farm of Sakha Agricultural Research Station, Kafrelsheikh on a 2.40 m^2^ plot area with three replicates in a completely randomized block design. Each plot area consists of four rows 2 m long and 30 cm apart. Grains were sown by hand-drilled at 400 seeds m^2^.

#### 2.5.1. Determination of Chlorophyll a and b

At the heading stage, a sample of flag leaves was randomly taken from each plot. Chlorophyll a and b contents in wheat leaves were measured using dimethyl sulfoxide [34]. Total chlorophyll was calculated and expressed as mg/g fresh weight (FW).

#### 2.5.2. Determination of Proline Content

Free proline content in the wheat leaves was determined according to Bates et al. [35]. A sample of 0.50 g leaves was homogenized in 5 mL of sulfosalicylic acid (3%) using a mortar and pestle. A 2 mL volume of the extract was taken in a test tube, and 2 mL of glacial acetic acid and 2 mL of ninhydrin reagent were added. The mixture was boiled in a water bath at 100 °C for 30 min. After cooling the reaction mixture, 6 mL of toluene was added and then transferred to a separating funnel. After thorough mixing, the chromosphere containing toluene was separated and absorption was read at 520 nm on a spectrophotometer. Toluene was used as a blank. The concentration of the proline was estimated by referring to a standard curve of proline.

### 2.6. Statistical Analysis

For the bioassay test, the LC_50_ value was estimated according to [36] via (Ldp line). The normality was checked by the Shapiro–Wilk normality test in the original data, which showed the normal distribution for the data. The data were analyzed using a one-way analysis of variance (ANOVA) and by using mortality percentages, the number of progeny/female, germination percentages, chlorophyll, and proline content (mg/g FW). The comparison between treatments was conducted by Tukey’s HSD post-hoc test at (*p* ≤ 0.05) via SPSS [37].

## 3. Results

### 3.1. Effect of Gamma Radiation on Adults S. oryzae

The data show that the mortality percentages of *S. oryzae* adults increased with increasing gamma radiation dose (Table 1 and Table 2). A significant difference in the mortality rate of the nonirradiated and irradiated adults at the different doses of gamma radiation and different time intervals was observed. At 48 h post-irradiation, the mortality rates were 5.50_,_ 19.80^,^ 21.00, and 48.8% for adults irradiated at 0.10, 0.25, 0.50, and 1.00 kGy, respectively. At the same doses, the mortality rates after 72 h of irradiation were 25.53, 38.86, 73.30, and 89.96%, respectively. Meanwhile, the mortality rates at 96 h post-irradiation ranged between 46.63 and 100%, and between 61.06 and 100% at 120 h post-irradiation. In addition, the mortality percentages of *S. oryzae* adults at 0.50 and 1.00 kGy were higher than 0.10 and 0.25 kGy after 72, 96, and 120 h of irradiation. On the contrary, no mortality of the nonirradiated *S. oryzae* adults occurred. The values of LD_50_ were 1.51, 0.22, 0.14, and 0.04 kGy at 48, 72, 96, and 120 h, respectively (Table 3).

### 3.2. Effect of Gamma Radiation on the Progeny of S. oryzae

The gamma radiation had a negative impact on the progeny, and there was no progeny noticed from *S. oryzae* adults after 60 days of irradiation treatment with all tested doses *F*5,18 = 34,848.60, *p* < 0.01 (Table 4).

### 3.3. Histological Changes in Irradiated and Non-Irradiated S. oryzae Adults

The histological alteration in adult insects of *S. oryzae* subjected to the highest tested dose (1.0 kGy) of gamma radiation was illustrated in Figure 1a,b. The abdominal cross-sections in nonirradiated *S. oryzae* adults (Figure 1a) show normal tissue with the epithelial layer, peritrophic membrane, and its muscles; furthermore, different layers, thin per trophic membrane, transitional epithelial cells, and gut lumen were normal. Comparably to the control, there were vacuoles in a section of the epithelium cell, the cell was completely dissolved and separated from the peritrophic membrane. The adjacent cell adhesion was lost slowly. The gut cells were atrophied and disorganized (Figure 1b).

### 3.4. The Germination Sensitivity to Gamma Radiation

The high doses of gamma radiation decreased the germination percentage when compared with control and the lower doses (*F*4,10 = 1586.26, *p* < 0.01). As illustrated in Table 5, the maximum germination percentage was recorded in the nonirradiated seedling. The final germination percentage was decreased with increasing the gamma radiation doses by 9% at 0.10 kGy, 20% at 0.25 kGy, 31% at 0.50 kGy, and 51% at 1.00 kGy. Statistically, there were no significant differences between 0.10 kGy of gamma dose and control treatment. Meanwhile, the exposure to gamma doses ≥0.25 kGy significantly decreased the germination percentage capacity at 7 days from exposure.

### 3.5. Effect of Gamma Irradiation on Chlorophyll (a and b) and Proline Contents of Wheat, T. vulgare Seedlings Leaves

Chlorophyll a and b contents were influenced by gamma irradiation. As shown in Table 6 and Table 7 the content of chlorophyll a was higher than chlorophyll b in both seedlings derived from grains exposed to gamma radiation and nonirradiated seedlings during both growing seasons. At doses of 0.50 and 1.00 kGy, chlorophyll a, b, and total chlorophyll concentrations were significantly lower in seedlings from irradiated grains than in the nonirradiated grains. During the first growing season at doses of 0.50 and 1.00 kGy, the chlorophyll a content was 5.32 and 3.76 mg/g FW, respectively compared with the nonirradiated seedlings (9.98 mg/g FW), and the high doses of gamma radiation decreased the chlorophyll a content compared with low doses (*F*4,10 = 21,186.22, *p* < 0.01). Similarly, Chlorophyll b contents at 0.50 and 1.00 kGy were 1.94 and 0.89 mg/g FW, respectively compared with seedlings from nonirradiated grains (2.55 mg/g FW) and the high gamma radiation doses had a negative impacts on chlorophyll a content (*F*4,10 = 3247.94, *p* < 0.01). A similar trend was observed during the second growing season (Table 7).

During the first growing season, gamma radiation at doses of 0.1 and 0.25 kGy had no significant impact on wheat proline content (0.25 and 0.26 mg/g FW), respectively as compared with the nonirradiated plants (0.23 mg/g FW). However, there was a significant increase in plant proline content at the higher doses of 0.5 and 1.0 kGy (0.30 and 0.36 mg/g FW), respectively compared with nonirradiated plants (Table 6) (*F*4,10 = 61.52, *p* < 0.01). A similar trend in proline content was observed during the second growing season (Table 7).

## 4. Discussion

The purpose of this study was to determine whether gamma irradiation exposure could limit the survival of *S. oryzae* and to assess the impact of exposure on seed germination. In the present study, the results showed that increasing radiation doses increased the mortality of *S. oryzae* adults. The dose of 1.00 kGy caused 100% mortality at 96 h post-irradiation. Abbas and Nouraddin [21] reported that a dosage of radiation 0.20 kGy caused 100% mortality of gamma-irradiated adult *T. castaneum* 28 days after treatment. Meanwhile, 0.10 kGy was the effective dose for 13–15 days old *T. castaneum* adults [20]. The dosage of 0.20 kGy caused 100% mortality of irradiated saw-toothed grain beetle *Oryzaephilus surinamensis* L. adults 28 days after treatment [38]. Gamma irradiation of adult *O. surinamensis* at a dosage range of 0.17–1.00 kGy results in absolute death 15 days after irradiation [39]. The difference between the present results and those from the previous studies is perhaps due to differences in the age of the experimental insects at the time of irradiation or differences in the experimental insect species.

The gamma radiation doses required to kill 50% (LD_50_) of *S. oryzae* were 1.51, 0.22, 0.14, and 0.04 kGy at 48, 72, 96, and 120 h, respectively. The results corroborate those of Ayvaz et al. [40], Silva and Arthur [41], and Farias et al. [42], who employed doses between 1.80 and 5.00 kGy to control various pests of stored grains, including *S. oryzae*, *Rhyzopersa dominica*, *T. castaneum*, and *T. confusum*.

Tilton and Brower [43] reported that curculionids were sensitive to irradiation. In the present study, we found that gamma radiation doses between 0.10 and 1.00 kGy caused complete sterility of 24 h old *S. oryzae* adults. Our findings agree with those of Follett et al. [44], who indicated that the treatment of adult weevils of *S. oryzae* with a radiation dose of 120 Gy resulted in no living adults after two weeks, showing that this dose of radiation caused adult death and sterility. In addition, Ignatowicz [45] found that the dose of 0.3 kGy of gamma radiation fully prevents the growth of immatures and sterilizes adults of stored-goods pest beetles.

Our results indicated that there was no progeny found from *S. oryzae* adults after 60 days of irradiation with the tested doses. These findings may be due to many reasons, such as the eggs of *S. oryzae* after radiation were unable to hatch into adults, and the adults after irradiation become sterile [46]. Moreover, many authors applied doses of irradiation to prevent the reproduction of stored-product pests. Sterilizing effects of gamma radiation on adult insects were studied at different doses. Brown et al. [47] found that 0.01 kGy was sufficient to cause sterility in adults of *S. granarius*. The gamma radiation at 0.07 kGy caused complete sterility in *T. castaneum* [48]. For three days, old *S. granarius* adults could not produce progeny when treated with a dosage of 0.10 kGy or above [49]. Adult females of the hide beetle *Dermestes maculatus* were sterile after gamma irradiation with 0.30 kGy [18]. Doses above 0.08 kGy produced a sterilizing effect on *S. oryzae* [50]. Moreover, the adults of *S. oryzae* cannot produce progeny when treated with a dosage of 0.10 kGy [51]. Gamma radiation can make an insect reproductively sterile by damaging the chromosomes of gonadal cells, specifically causing germ-cell chromosome fragmentation (dominant lethal mutations, translocations, and other chromosomal aberrations), which leads to the production of imbalanced gametes and subsequently the inhibition of mitosis and death of fertilized eggs or embryos [52]. Furthermore, the gamma radiation had a strong effect on the abdominal sections of *S. oryzae* adults after 48 h of gamma irradiation at 1.00 kGy. These findings agreed with Ahmadi et al. [53], who indicated that the genital cells of *S. oryzae* were more sensitive to irradiation exposure and correlated with its high mortality percentage than *Tribolium castaneum* and *Callosobruchus maculatus*.

In the current study, the content of chlorophyll a was higher than chlorophyll b in both plants derived from seeds exposed to gamma radiation and control plants during both growing seasons. Seedlings from grains exposed to gamma radiation (0.50 and 1.00 kGy) exhibited lower chlorophyll a and b levels. Data presented herein are in agreement with the results of previous reports. For instance, Marcu et al. [28] showed that chlorophyll a and b contents significantly decreased in maize, *Zea mays* leaves derived from irradiated grains at 0.30 kGy than in control grains. Chloroplasts are sensitive to gamma radiation compared with other cells, especially thylakoids being heavily swollen [54]. The reduction in chlorophyll b is due to the destruction of chlorophyll b biosynthesis by gamma radiation or degradation of chlorophyll b precursors [26]. Increasing growth occurred as a result of the modification in photosynthesis by irradiated plants [54,55].

Proline content was considerably increased by increasing gamma irradiation dosages, and these findings agreed with previous studies that indicated a positive correlation between proline accumulation and enhancing stress tolerance in the plants [29,30,31]. Proline is a source of energy, carbon, and nitrogen for the recovering tissues. Accumulation of proline primarily occurs in response to various abiotic and biotic stresses that cause dehydration of the plant tissue [56]. Therefore, it can be suggested that proline accumulation can protect plant cells against damage induced by gamma radiation. Data presented herein revealed a significant increase in plant proline content at the higher doses of 0.50 and 1.00 kGy compared with nonirradiated plants. Relatively similar results were observed by other researchers, e.g., Borzouel et al. [27] found that the proline content of wheat seedlings irradiated at 0.10 kGy contained the highest amount of proline (1.71 mg/g FW), whereas only 0.92 mg/g FW of proline was detected in nonirradiated seedlings. 

## 5. Conclusions

It could be concluded that the mortality percentages of *S. oryzae* adults enhanced with increasing gamma radiation doses. The 1.51 kGy dose of gamma radiation killed 50% of *S. oryzae* adults after 48 h, and the gamma radiation dose of 1.00 kGy killed 100% of *S. oryzae* adults after 96 h of exposure. Moreover, the gamma radiation enhanced the sterility of *S. oryzae* and all tested doses caused complete sterility to 24 h old adults, which could be considered a promising direction for protecting stored grains. A histological alteration was found at a gamma radiation dose of 1.00 kGy. Additionally, plant proline increased at the higher gamma radiation doses of 0.5 and 1.0 kGy. The total chlorophyll concentrations were lower in plants from irradiated grains than in nonirradiated grains. Finally, gamma radiation can be used as an eco-friendly method to control stored-product pests. 

## Figures and Tables

**Figure 1 biology-11-01295-f001:**
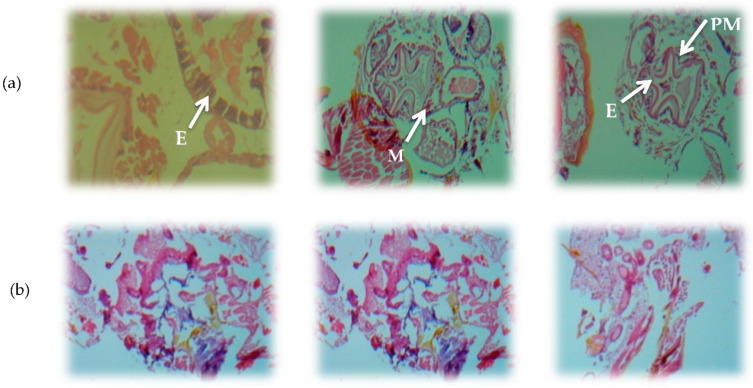
(**a**) Abdominal sections in the nonirradiated *S. oryzae* adults (control). (**b**) Abdominal sections in the *S. oryzae* adults after 48 h of gamma irradiation at 1.00 kGy. PM = Peritrophic membrane, M = Muscles, E = Epithelial cells.

**Table 1 biology-11-01295-t001:** Mortality percentage of *S. oryzae* adults after exposure to different doses of gamma radiation at different times.

Gamma Radiation Dose (kGy)	Time (h) after Irradiation			
	48	72	96	120
0.00 (control)	0.00 ± 0.00 d	0.00 ± 0.00 c	0.00 ± 0.00 c	0.00 ± 0.00 c
0.10	5.5 ± 2.1 c	25.53 ± 2.93 b	46.63 ± 6.60 b	61.06 ±5.53 b
0.25	19.8 ± 1.8 b	38.86 ± 2.94 b	53.30 ± 5.09 b	67.76 ±7.76 b
0.50	21.0 ± 0.1 b	73.30 ± 5.77 a	79.96 ± 8.81 a	97.76 ±2.23 a
1.00	48.8 ± 4.8 a	89.96 ± 3.83 a	100 ±0.00 a	100 ±0.00 a

Values are the mean ± S.E. The means of each column followed by the different letters are significantly different at 0.05 level according to Tukey’s multiple range test.

**Table 2 biology-11-01295-t002:** Analysis of variance of mortality percentage of *S. oryzae* after exposure to different doses of gamma radiation at different times.

Variable	Time (h) after Irradiation			
	48	72	96	120
SS	5731.50	20,975.70	22,935.30	26,178.00
MS	1432.87	5243.92	5733.82	6544.50
F value	3306.63	14,982.64	10,118.51	17,072.60
*p*-value	<0.01	<0.01	<0.01	<0.01

Variation source = gamma radiation doses, degree of freedom = 5, SS = sum of squares, MS = mean squares.

**Table 3 biology-11-01295-t003:** Lethal effect of gamma radiation on *S. oryzae* at different times (h).

Time (h) after Irradiation	LD_50_ (kGy)	Confidence Limit		Slope	Chi-Square X^2^	*p*-Value
		Lower	Upper			
48.00	1.51	0.85	1.73	1.08 ± 0.15	6.16	0.11
72.00	0.22	0.19	0.26	1.16 ± 0.14	3.88	0.01
96.00	0.14	0.10	0.16	1.72 ± 0.15	3.65	1.30
120.00	0.04	0.03	0.07	1.28 ± 0.17	4.85	0.18

**Table 4 biology-11-01295-t004:** Effect of different doses of gamma radiation on progeny production of *S. oryzae* after 60 days of irradiation.

Gamma Radiation Dose (kGy)	No. Progeny/Female
Control	241.00 ± 3.46 a
0.10	00.00 ± 0.00 b
0.25	00.00 ± 0.00 b
0.50	00.00 ± 0.00 b
1.00	00.00 ± 0.00 b

Values are the mean ± S.E. The means of each column followed by the different letters are significantly different at 0.05 level according to Tukey’s multiple range test.

**Table 5 biology-11-01295-t005:** Effects of gamma irradiation on germination (%) after 7 days post-treatment.

Gamma Dose (kGy)	Germination (%)
Control	98.34 ± 1.66 a
0.10	90.01 ± 2.88 a
0.25	73.34 ± 1.66 b
0.50	61.68 ± 1.66 c
1.00	46.88 ± 1.66 d

Values are the mean ± S.E. The means of each column followed by the different letters are significantly different at 0.05 level according to Tukey’s multiple range test.

**Table 6 biology-11-01295-t006:** Effect of gamma irradiation on chlorophyll (a and b) and total proline content of wheat, *T. vulgare* leaves in 2019/2020 growing season.

Gamma Dose (kGy)	Chlorophyll Content (mg/g FW)			Proline (mg/g FW)
	*A*	*B*	Total	
Control	9.98 ± 0.03 c	2.55 ± 0.08 c	12.53	0.23 ± 0.00 c
0.10	10.56 ± 0.05 a	3.01 ± 0.07 a	13.57	0.25 ± 0.00 c
0.25	10.03 ± 0.07 b	2.89 ± 0.01 b	12.92	0.26 ± 0.01 c
0.50	5.32 ± 0.01 d	1.94 ± 0.32 d	7.26	0.30 ± 0.08 b
1.00	3.76 ± 0.03 e	0.89 ± 0.02 e	4.65	0.36 ± 0.08 a

Values are the mean ± S.E. The means of each row followed by the different letters are significantly different at 0.05 levels according to Tukey’s multiple range test.

**Table 7 biology-11-01295-t007:** Effect of gamma irradiation on chlorophyll (a and b) and total proline content of wheat, *T. vulgare* leaves in 2020/2021 growing season.

Gamma Dose (kGy)	Chlorophyll Content (mg/g FW)			Proline (mg/g FW)
	*A*	*B*	Total	
Control	9.98 ± 0.03 c	2.56 ± 0.08 c	12.54	0.23 ± 0.00 c
0.10	11.04 ± 0.05 a	3.01 ± 0.07 a	14.05	0.25 ± 0.00 c
0.25	10.35 ± 0.07 b	3.05 ± 0.01 b	13.40	0.27 ± 0.01 c
0.50	6.64 ± 0.01 d	2.11 ± 0.32 d	8.75	0.30 ± 0.07 b
1.00	4.71 ± 0.03 e	1.16 ± 0.02 e	5.87	0.37 ± 0.08 a

Values are the mean ± S.E. The means of each row followed by the different letters are significantly different at 0.05 levels according to Tukey’s multiple range test.

## Data Availability

All data generated or analyzed during this study are included in this published article.

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
