# Peer review of "Gamma Radiation: An Eco-Friendly Control Method for the Rice Weevil, Sitophilus oryzae (L.) (Coleoptera: Curculionidae)"

_biology, 2022, doi:10.3390/biology11091295_

Round 1

Reviewer 1 Report

The authors carried out a study to evaluate the percentage of mortality, sterility, and histological damage in adults of the rice weevil, Sitophilus oryzae, exposed to gamma radiation at different doses and times. They also determined germination, chlorophyll, and proline content in wheat seedlings from grains treated by the same method.  In this study, the authors present some important findings. However, I have some minor comments that should be addressed to improve the quality of this article. 

Title

Line 3: The correct species name is Sitophilus oryzae. Please, correct it here and elsewhere in the manuscript.

Abstract

Line 26: "histological alternation"'? Perhaps the authors meant histological alteration.

Keywords

Line 33. Gamma radiation and Sitophilus oryzae are in the title; please change these keywords.

Introduction

Lines 52-54: It would be more suitable "Also, El-Nabawy et al. [10,11] showed that using flowering plants"

Materials and methods

Lines 93-95: It would be better if the authors support this and other parts of the methods with references (101-108, 114-121, etc.).

Results

Lines 171-177: The authors do not provide Tukey test results and P-values for observation times (48-120 h) for mortality rate after radiation exposure. It's important to aggregate the results of Tukey's test and the P-value here and elsewhere in the manuscript (lines 195-196, 219-222, 235-254).

Lines 198-199: The statement is confusing. What do the authors mean by histological alternation? Alternation is not synonymous with damage or alteration. Please clarify.

Line 203: peritrophic membrane

Lines 283-291: I suggest authors rewrite this part to make it clear for readers and better connect with the explanation of gamma radiation damage.

Line 299: T. castaneum

Conclusion

No comments.

References

Lines 363-364: Genus and species scientific names should be in italics, here and elsewhere in the references (377-378, 430-432).  

Author Response

Dear Reviewer 1,

Thank you for your precious time in reviewing our manuscript and providing valuable and great comments. It was your valuable and insightful comments that led to possible improvements in the current version. Authors have carefully considered the comments and tried our best to address every one of them. We hope the manuscript after careful revisions meet your high standards. The authors welcome further constructive comments if any. Below we provide the point-by-point responses for all reviewers’ comments.

Reviewer 1, comments and Suggestions for Authors

The authors would like to thank you for your valuable and great comments. It was your valuable and insightful comments that led to possible improvements in the current version. All corrections were done according to your advices.

Comment 1: Line 3: The correct species name is Sitophilus oryzae. Please, correct it here and elsewhere in the manuscript.

Response 1:  Thank you very much for your great comment and your advice: we changed the it to Sitophilus oryzae  according to your advice as follow.

Comment 2  Line 26: "histological alternation"'? Perhaps the authors meant histological alteration.

Response 2:  Thank you very much for your comment, we changed alternation to alteration.

Comment 3: Line 33. Gamma radiation and Sitophilus oryzae are in the title; please change these keywords..

Response 3: Thank you very much for your questions. We changed the key words to be Sterility; chlorophyll; proline; progeny; germination.

Comment 4: Lines 52-54: It would be more suitable "Also, El-Nabawy et al. [10,11] showed that using flowering plants"

Response 4: Thank you very much for your advice; we changed it according to your advice to be Also, El-Nabawy et al. [10,11] showed that using flowering plants

Comment 5 Lines 171-177: The authors do not provide Tukey test results and P-values for observation times (48-120 h) for mortality rate after radiation exposure. It's important to aggregate the results of Tukey's test and the P-value here and elsewhere in the manuscript (lines 195-196, 219-222, 235-254).

Response 5: Thank you very much for your comment we changed all the manuscript according to your great advice and we added the results of statistical analysis.  

Comment 6: Lines 198-199: The statement is confusing. What do the authors mean by histological alternation? Alternation is not synonymous with damage or alteration. Please clarify.

Response 6: Thank you very much for your comment it was a mistake by us; we changed it to be alteration

Comment 7: Line 203: peritrophic membrane

Response 7: Thank you very much for your advice; we changed it to be peritrophic membrane.

Comment 8: Lines 283-291: I suggest authors rewrite this part to make it clear for readers and better connect with the explanation of gamma radiation damage.

Response 8: Thank you very much for your advice; we changed this part to be more obvious according to your recommendation. Our results indicated there was no progeny found from S. oryzae adults after 60 days of irradiation with the tested doses. These findings may be because of many reasons such as the eggs of S. oryzae after radiation were unable to hatch into adults and the adults after irradiation become sterile Salehi [2005]. Also, many authors applied doses of irradiation to prevent the reproduction of stored product pests. Sterilizing effects of gamma radiation on adult insects were studied at different doses. Brown et al. [38] found that 0.01 kGy was sufficient to cause sterility in adults of S. granarius. The gamma radiation at 0.07 kGy caused complete sterility in T. castaneum [39]. For three days old S. granarius adults could not produce progeny when treated with a dosage of 0.10 kGy or above [40]. Adult females of the hide beetle Dermestes maculatus were sterile after gamma irradiation with 0.30 kGy [16]. Doses above 0.08 kGy produced a sterilizing effect on S. oryzae [41]. Also, the adults of S. oryzae cannot produce progeny when treated with a dosage of 0.10 kGy [42].  

Comment 9: Line 299: T. castaneum

Response 9: Thank you very much for your comment; we changed it to be T. castaneum

Reviewer 2 Report

August 03 2022

Journal of Biology

Assistant Editor, MDPI Romania

Dear Authors,

Attached you will find my comments and suggestions about the manuscript Gamma radiation: an eco-friendly control method for the rice weevil, Sitophilus orayzae (L.) (Coleoptera: Curculionidae)” (biology-1827691-v1), written by Nasr et al.

General comments

P1; L3:- Is correct this name Sitophilus orayzae or Sitophilus oryzae.

P2, L67: Sitotroga cerealella, change to italics.

P3; L93:- Please, add the dimensions of containers and change to “glass container”.

P3; L98:- Change “estimation of progeny production” to “effects on reproduction”.

P3; L111:- Change “jars” by “containers”.

P3; Ll01-102:- Indicate or describe why these doses were applied?

P4; L124:- Change to “Effect of gamma irradiation on germination”.

P4, L136:- Change “investigation” by “phase”,

P4 and 5; L160 and 164:- In the Statistical analysis section. The authors must describe the variables evaluated in each experimental phase and should be mention what type of experimental design tested or used. Also is necessary include how many experimental units were in each phase. In the results section is also necessary to include the statistics (P; t, df; F).

P5; L182:- How many insects were treated per dose. Add N = and the number of insects per treatment.

P9; L266:- Delete “In relation to the dosage of radiation”, 

P9; L267:- Add “dosage of radiation”

Other commentary

Irradiation is used for various purposes (SIT, post-harvest quarantine treatment, resistance induction in plants, seed pest control, etc.). It is necessary to align the introduction to the objective of the paper.

Author Response

Reviewer 2, comments and Suggestions for Authors

The authors would like to thank you very much for your valuable and great comments. It was your valuable and insightful comments that led to possible improvements in the current version. All corrections were done according to your advices.

General comments

  • P2, L67: Sitotroga cerealella, change to italics.

Response: Thank you very much for your comment we changed it to italic.

  • P3; L93:- Please, add the dimensions of containers and change to “glass container”

Response: Thank you very much for your advice we changed it to glass containers

  • P3; L98:- Change “estimation of progeny production” to “effects on reproduction”.

 Response: Thank you very much for your great comment and advices we changed “estimation of progeny production” to “effects on reproduction”.

  • P3; L111:- Change “jars” by “containers”.

Response: Thank you very much for your great comment and advices we changed “jars” to “containers”.

  • P3; Ll01-102:- Indicate or describe why these doses were applied?

Response: Thank you very much for your comment and advices we used these doses according to the maximum and minimum of previous studies on S. oryzae

  • P4; L124:- Change to “Effect of gamma irradiation on germination”.

Response: Thank you very much for your comment we changed that it to be more suitable according to your advice 

  • P4, L136:- Change “investigation” by “phase”

Response: Thank you very much for your comment we changed “investigation” by “phase” according to your advice

  • P4 and 5; L160 and 164:- In the Statistical analysis section. The authors must describe the variables evaluated in each experimental phase and should be mention what type of experimental design tested or used. Also is necessary include how many experimental units were in each phase. In the results section is also necessary to include the statistics (P; t, df; F).

Response: Thank you very much for your comment we added the detailed information about statistical analysis according to your recommendation.

  • P5; L182:- How many insects were treated per dose. Add N = and the number of insects per treatment.

Response: Thank you very much for your comment the numbers of insects were fifty unsexed newly emerged adult. Also, four replicates were used for each dose level and control.

  • P9; L266:- Delete “In relation to the dosage of radiation”, 

Response: Thank you very much for your comment we deleted In relation to the dosage of radiation.

  • P9; L267:- Add “dosage of radiation”

Response: Thank you very much for your advice we added “dosage of radiation”

Reviewer 3 Report

Dear Authors,

the article provides interesting information on the effects of various doses of gamma rays on survival of S. oryzae and regarding certain biological parameters related to wheat grains that are co-irradiated with the pest. However, the data regarding the pest are not completely new and certain relevant studies on the topic have not been cited. The neglected existing literature reduces the novelty of the experimental work. Also, discussion does not make it clear how the obtained data can support the use of gamma irradiation to sanitize wheat. Is the objective of the strategy the sanification of stored wheat? How to irradiate large amounts of wheat? How to assure an approximately constant dose to all grains? Focusing on these points would significantly enhance the value of the discussion to support conclusions. Also, language requires a substantial revision because there are several mistakes regarding grammar. In my opinion, the article is not suitable for publication in present form as it requires major revisions but I encourage Authors to improve it.

Specific comments:

Title: “orayzae” has to be revised to “oryzae”

line 18: “orayzae” has to be revised to “oryzae”

line 92: please add country

line 111: “After that emerged S. oryzae adults were counted.” At this point, it is not clear. Did the irradiated adults produce eggs on the grains? have been all the treated adults removed? have been the produced eggs counted? what do you mean for "after that emerged adults were counted"? do you mean that you counted all adults emerged after 60 days? since which event the count of 60 days has started?

Line 139: methods of irradiation could be described once

Line 145 and line 149: please, make it clear, here or in introduction, why chlorophyll and proline have selected.

Line 159 (statistical analysis): It is not fully clear. It would be preferrable to specify which kind of data have been subjected to ANOVA. Have been percent data transformed before analysis?

Line 197: In my opinion, a single paragraph comparing control and irradiated samples would be enough. Also figure 1 and 2 should be combined ina single panel to allow for a easier comparison between treatments. The description of the effects should be improved and figures should report graphical marks common to treatment and control to highlight differences.

Line 226: Table 4. It is not necessary to also highlight percent reduction

Line 249: “During the first growing season…” It is not clear to me why the grains of two different growing season have been irradiated. Are they expected to be different?

Line 314: the hypothesis regarding proline is not supported by data. It could be assumed that proline content is increased by the irradiation but nothing about its role in this specific case has been demonstrated.

Conclusions: first sentence is not clear. In general, conclusions are not sufficiently supported by data and mainly repeat the findings of the experimentation. Results should be better discussed in the context of the methods used to control this pest and taking into account feasibility, costs, and benefits of the proposed strategy.

Below, I list references that should be read and cited by the authors:

Cornwell PB 1966.  Susceptibility of the grain and rice weevil Sitophilus granarius and Sitophilus zeamais Mots. to gamma radiation. In: Cornwell PB ed., The Entomology of Radiation Disinfestation of Grain. Pergamon press Ltd. 1966. Pag 1-18.

Follet et al., 2013. Irradiation quarantine treatment for control of Sitophilus oryzae (Coleoptera: Curculionidae) in rice. Journal of Stored Products Research 52: 63-67.

Tunçbilek, AÅž. 2005. Effect of 60Co gamma radiation on the rice weevil, Sitophilus oryzae (L.). Anzeiger für Schädlingskunde, Pflanzenschutz, Umweltschutz 68: 37-38.

Ignatowicz S. Irradiation as an alternative to methyl bromide fumigation of agricultural commodities infested with quarantine stored product pests. Irradiation as a phytosanitary treatment of food and agricultural commodities. 2004:51–66.

Franco SSH, Franco JG, Arthur V, 1997. Determinacao da dose esterilizante de radiacao gama do Cobalto-60 para Sitophilus oryzae, S. zeamais, S. granarius, em arroz, milho e trigo. Ecossistema, 22:120-121

Machi et al., 2017. Ionizing Radiation and the Influence of Package to Control of Sitophilus oryzae in Rice. Australian Journal of Basic and Applied Sciences, 11(15) December 2017, Pages: 71-75

Salehi L., Ranjbar Kaboutarkhani S. The Effects Of 60Co Gamma Radiation On Develomental Stages And Fecundity Of The Rice Weevile, Sitophilus Oryzae L. (Col: Curculionidae) Journal Of Agricultural Sciences , FALL 2005, 11(3).

Author Response

Reviewer 3, comments and Suggestions for Authors

The authors would like to thank you very much for your valuable and great comments. It was your valuable and insightful comments that led to possible improvements in the current version. The corrections were done according to your advice.

Comment 1: Title: “orayzae” has to be revised to “oryzae”

Response 1: Thank you very much for your comment it as a mistake from us and we changed it be “oryzae”

Comment 2: line 18: “orayzae” has to be revised to “oryzae”

Response 2: Thank you very much for your advice it as a mistake from us and we changed it be “oryzae”

Comment 3: line 92: please add country

Response 3: Thank you very much for your comment: we added the country in the following paragraph  Irradiation of S. oryzae was conducted at an irradiator facility with 60Co gamma source at a dosage rate of 0.930 kGy / hr., at the National Center for Radiation Research and Technology, Atomic Energy Authority, Cairo, Egypt.

Comment 4: line 111: “After that emerged S. oryzae adults were counted.” At this point, it is not clear. Did the irradiated adults produce eggs on the grains? have been all the treated adults removed? have been the produced eggs counted? what do you mean for "after that emerged adults were counted"? do you mean that you counted all adults emerged after 60 days? since which event the count of 60 days has started?

Response 4: Thank you very much for your suggestion we changed the sentences to more clear according to your recommendation. Fifty unsexed newly emerged adult weevils of S. oryzae were randomly selected from the stock culture by sieving and were transferred to Petri dishes containing 20 g wheat (variety Giza 171) grains as food and then exposed to gamma irradiation with the previous concentrations. A control treatment was subjected to the same conditions without radiation exposure. Immediately after treatment, irradiated and non-irradiated weevils were returned to the laboratory, and Petri dishes were placed in an incubator at a temperature of 28 ± 1°C and 65 ± 5% RH. Four replicates were used for each dose level and control. Adult weevil mortality was recorded after 48, 72, 96, and 120 hrs, and dead insects were removed. The median lethal dose (LD50) of gamma rays with a 95% confidence limit was estimated by probit analysis. All containers were returned to the incubator for 60 days. After 60 days of irradiation, all emerged S. oryzae adults were counted and recorded.

Comment 5: Line 139: methods of irradiation could be described once

Response 5: Thank you very much for your comment  and we described the methods of irradiation once and deleted other senteces according to your advice.

Comment 6: Line 145 and line 149: please, make it clear, here or in introduction, why chlorophyll and proline have selected.

Response 6: Thank you very much for your comment we revised it and we added Sadeghipour [2020], Elewa et al. [2017] and Qirat et al. [2018] showed a positive correlation between proline accumulation and enhancing stress tolerance in the plants. Also, AlKahtani et al. [2021] showed that one of the most crucial factors for evaluating the environmental stress on plants is the chlorophyll content of the leaves, which declines under pressure.

Our objective was to evaluate the effect of different doses of gamma irradiation on the mortality percentages and the progeny production of S. oryzae. The study was also designed to assess the effect of gamma irradiation (1.0 kGy) on the cellular organization and other tissues of S. oryzae adults. Additionally, to study the effects of gamma radiation on germination, chlorophyll and proline contents of wheat seedlings to estimate the use of irradiated grains in farming.

Comment 7: Line 159 (statistical analysis): It is not fully clear. It would be preferrable to specify which kind of data have been subjected to ANOVA. Have been percent data transformed before analysis?

Response 7: Thank you very much for your comment we added the detailed information about statistical analysis according to your recommendation.

Comment 8: Line 197: In my opinion, a single paragraph comparing control and irradiated samples would be enough. Also figure 1 and 2 should be combined ina single panel to allow for a easier comparison between treatments. The description of the effects should be improved and figures should report graphical marks common to treatment and control to highlight differences.

Response 9: Thank you very much for your advice and we combined the control with the irradiation treatment as well as in the figures 1, and 2 as shown in the following paragraph. The histological alteration in adult insects of S. oryzae subjected to the highest tested dose (1.0 kGy) of gamma radiation was illustrated in Figures (1 a and b). The abdominal cross-sections in non-irradiated S. oryzae adults (Figure 1. (a)) show normal tissue with the epithelial layer, peritrophic membrane, and its muscles, also, different layers, thin per trophic membrane, transitional epithelial cells, and gut lumen were normal. Comparably to the control, there were vacuoles in a section of the epithelium cell, the cell was completely dissolved and separated from the peritrophic membrane. The adjacent cell adhesion was lost slowly. The gut cells were atrophied and disorganized (Figure 1. (b)).

Comment 9: Line 226: Table 4. It is not necessary to also highlight percent reduction

Response 9: Thank you very much for your comment and we deleted the  percent reduction according to your  advice

Comment 10: Line 249: “During the first growing season…” It is not clear to me why the grains of two different growing season have been irradiated. Are they expected to be different?

Response 10: Thank you very much for your advice and we did 2 growing season to confirm the field experiments.  Also because field experiments may many different factors such as weather changes and so on. Thus we did 2 growing season.

Comment 11: Line 314: the hypothesis regarding proline is not supported by data. It could be assumed that proline content is increased by the irradiation but nothing about its role in this specific case has been demonstrated.

Response 11: Thank you very much for your advice and we added some previous studies to indicate the effect of radiation on proline content. Proline content was considerably increased by increasing gamma irradiation dosages and these findings agreed with previous studies that indicated a positive correlation between proline accumulation and enhancing stress tolerance in the plants Sadeghipour [2020], Elewa et al. [2017] and Qirat et al. [2018]. Proline is a source of energy, carbon, and nitrogen for the recovering tissues. Accumulation of proline primarily occurs in response to various abiotic and biotic stresses that cause dehydration of the plant tissue [47]. Therefore, it can be suggested that proline accumulation can protect plant cells against damage induced by gamma radiation. Data presented herein revealed a significant increase in plant proline content at the higher doses of 0.50 and 1.00 kGy compared with non-irradiated plants. Relatively similar results were observed by other researchers, e.g., Borzouel et al. [25] found that the proline content of wheat seedlings irradiated at 0.10 kGy contained the highest amount of proline (1.71 mg/g FW), whereas only 0.92 mg/g FW of proline was detected in non-irradiated seedlings.

Comment 12: Conclusions: first sentence is not clear. In general, conclusions are not sufficiently supported by data and mainly repeat the findings of the experimentation. Results should be better discussed in the context of the methods used to control this pest and taking into account feasibility, costs, and benefits of the proposed strategy.

Response 12: Thank you very much for your comment and we changed it to be more clear according to your recommendation as in the following paragraph. It could be concluded that the mortality percentages of S. oryzae adults enhanced with increasing gamma radiation doses. The gamma radiation enhanced the sterility of S. oryzae and it could be considered a promising direction for protecting stored grains. Additionally, plant proline increased at the higher gamma radiation doses of 0.5 and 1.0 kGy. The total chlorophyll concentrations were lower in plants from irradiated grains than in the non-irradiated ones. Finally, gamma radiation can be used as an eco-friendly method to control stored-product pests. Also, we recommend the farmers to use the wheat grains without irradiation for farming.

Round 2

Reviewer 3 Report

Dear Authors,

the article has been substantially improved. I just have doubts about novelty but overall the article can be published. I am adding a few further suggestions.

 In figure 1, the scale of the two series of pictures does not seem to be the same. I suggest to add a scale bar and also arrows in the pictures regarding treated samples to highlight the same structures of control, allowing for an easier comparison.

 Conclusions could be improved as they mainly repeat the findings of the experimentation but a preliminary discussion regarding the context and feasibility, costs, and benefits of the proposed strategy should be added before writing that latter is an actual perspective.

The sentence “Moreover, we recommend that farmers use the wheat grains without irradiation for farming” seems to me unnecessary.

Author Response

The authors would like to thank you very much for your valuable and great comments. It was your valuable and insightful comments that led to possible improvements in the current version. The corrections were done according to your advice.